# MALDI Imaging, a Powerful Multiplex Approach to Decipher Intratumoral Heterogeneity: Combined Hepato-Cholangiocarcinomas as Proof of Concept

**DOI:** 10.3390/cancers15072143

**Published:** 2023-04-04

**Authors:** Elia Gigante, Hélène Cazier, Miguel Albuquerque, Samira Laouirem, Aurélie Beaufrère, Valérie Paradis

**Affiliations:** 1Centre de Recherche sur L'inflammation, Inserm, Université Paris Cité, F-75018 Paris, France; 2Service d’Hépato-Gastroentérologie et Cancérologie Digestive, Hôpital Robert Debré, F-51090 Reims, France; 3Plateforme iMAP, Centre de Recherche sur L'inflammation, Inserm, Université Paris Cité, F-75018 Paris, France; 4Département de Pathologie, Assistance Publique-Hôpitaux de Paris, FHU MOSAIC, Hôpital Beaujon, F-92110 Clichy, France

**Keywords:** combined hepato-cholangiocarcinomas, diagnosis, intratumoral heterogeneity, MALDI imaging, immunohistochemistry, in silico digestion

## Abstract

**Simple Summary:**

To provide a comprehensive molecular characterization of combined hepatocholangiocarcinomas (cHCC-CCA), the MALDI (Matrix Assisted Laser Desorption Ionization) imaging approach was used to acquire high-resolution spatialized molecular profiles directly from fixed tissue sections without a previous labeling step. The heterogeneity of cHCC-CCA was explored through the analysis of whole tumor slides by comparing peptidic profiles and histological and immunophenotypical images. MALDI imaging analysis was able to identify both tumor components (HCC and iCCA) and also small tumor areas not seen at the microscopic level. In addition, the morphological distribution of tryptic peptides of the different immunophenotypic markers was analyzed in silico. Their distribution allowed us to identify areas initially unlabeled by immunohistochemistry (IHC). This study demonstrates that an in silico MALDI imaging approach may improve conventional histological analysis and allow the development of new markers.

**Abstract:**

Combined hepato-cholangiocarcinomas (cHCC-CCA) belong to the spectrum of primary liver carcinomas, which include hepatocellular carcinomas (HCC) and intrahepatic cholangiocarcinomas (iCCA) at both ends of the spectrum. Mainly due to the high intratumor heterogeneity of cHCC-CCA, its diagnosis and pathological description remain challenging. Taking advantage of in situ non-targeted molecular mapping provided by MALDI (Matrix Assisted Laser Desorption Ionization) imaging, we sought to develop a multiscale and multiparametric morphological approach, integrating molecular and conventional pathological analysis. MALDI imaging was applied to five representative cases of resected cHCC-CCA. Principal component analysis and segmentations with MALDI imaging techniques identified areas related to either iCCA or HCC and also hidden tumor areas not visible microscopically. In addition, the overlap between MALDI segmentation and immunostaining provided a comprehensive description of cHCC-CCA tumor heterogeneity by identifying transitional and micro-metastatic areas. Moreover, a list of peptides derived from in silico digestion was obtained for each immunohistochemical marker and was matched within the peptide peak list acquired by MALDI. Comparison of immunostaining images with ions from in silico digestion revealed an accurate identification of iCCA and HCC areas. Our study provides further evidence on the performance of MALDI imaging in exploring intratumor heterogeneity and offering virtual multiplex immunostaining through a single acquisition.

## 1. Introduction

Combined hepato-cholangiocarcinoma (cHCC-CCA) is a rare primary liver carcinoma (PLC) compared to hepatocellular carcinoma (HCC) and intrahepatic cholangiocarcinoma (iCCA). A cHCC-CCA diagnosis is histologically based, identifying both HCC and iCCA components within the same tumor [1,2]. The frequency of cHCC-CCA is estimated to be between 2–5% of all PLCs; this rate is probably underestimated because of the difficulty of diagnosing this cancer [3,4]. Several morphological classifications have been proposed, describing different subtypes, leading to a certain degree of confusion and misunderstanding. The latest classification has been simplified and recognizes as cHCC-CCA only those lesions in which HCC and CCA areas are present, based on routine hematin-eosin (HE) staining. Also, it is important to note that cHCC-CCA diagnosis is proposed whatever their respective proportion [2]. To support the morphological diagnosis, some additional immunohistochemical (IHC) markers reflecting cell differentiation (hepatocytic and cholangiocytic) as well as stem cell phenotype are used; these latter are useful to better characterize PLC and to guide differential diagnoses without being diagnostic by themselves [2,3,4]. Recently, Nestin, a marker for cHCC-CCA, has also been reported to have prognostic value [5]. In the field of liver oncogenesis, cHCC-CCA illustrates the best example of morphological intratumor heterogeneity. Despite such morphological intratumor heterogeneity, several studies demonstrated their clonal origin and reported *TP53*, *TERT*, and *PTEN* as the most frequent genetic alterations identified [6,7]. In addition, about 25% of cHCC-CCAs had known potentially targetable alterations such as *BRCA2*, *ERBB2*, *IDH1*, *BRAF*, *FGFR2*, and *MET* [8]. Most of these genetic alterations are described either in HCC or in iCCA.

Intratumor heterogeneity is a challenging issue in terms of diagnosis that may impact patient management and clinical outcome. To provide a comprehensive molecular overview of highly heterogeneous tumors, such as cHCC-CCA, in situ approaches are the most appropriate, providing molecular characterization with respect to tumor spatial organization that cannot be achieved by microdissection techniques. MALDI imaging has emerged as a powerful technique for analyzing the spatial arrangement of various kinds of molecules (proteins, peptides, lipids, and metabolites) directly from tissue sections without preliminary microdissection and labelling steps [9]. Mass Spectrometry Imaging (MSI) has already made it possible to identify novel diagnostic and prognostic markers in several types of human malignancies, including HCC and iCCA; once identified, candidate markers can then be analyzed via routine techniques such as immunohistochemistry [9,10,11,12,13]. Interestingly, MSI has made it possible to correlate intratumor heterogeneity with a worse prognosis in several cancers, such as gastric carcinoma [14]. Considering that cHCC-CCA is by definition a tumor with a high level of heterogeneity, MSI appears the most appropriate approach to investigate this PLC and provides further molecular tools [14].

The aims of our study were to unravel the intratumoral heterogeneity of cHCC-CCA using MSI and to evaluate its potential to better characterize this tumor.

## 2. Materials and Methods

### 2.1. cHCC-CCA Cases Selection

Five cHCC-CCA cases were retrieved from our pathology department from among archival surgical specimens. The cases were selected according to their typical morphological features with well-defined areas of HCC and iCCA, as defined by the 2019 WHO classification [3]. They were examined by two pathologists who specialized in liver disease (AB, VP) using Hematein Eosin Safran (HES) staining (Appendix A). The most representative tumor sample section for each case was selected for MSI and immunohistochemical analysis.

Immunohistochemical analysis included a panel of 6 antibodies reflecting cell differentiation [hepatocellular (HepPar-1 and Glypican-3), cholangiocytic (CK 7 and CK 19)] and stem cell phenotype (EpCAM and Nestin) (Appendix A). An automated immunohistochemical stainer was used according to the manufacturer’s guidelines (streptavidin-peroxidase protocol; Benchmark, Ventana, Tucson, AZ, USA). Immunostained slides were digitalized (ScanScope AT turbo**^®^**, Leica, Wetzlar, Germany) at ×20 magnification.

### 2.2. MALDI Imaging

#### Sample Treatment and Acquisition

For all cases, analysis was performed on one 3-µm-thick serial section mounted on ITO (Indium Tin Oxide)-coated conductive slides for MALDI imaging analysis. Slides were dried at 37 °C overnight before analysis. For MSI analysis, slides were pre-heated at 85 °C for 15 min and dewaxing was done with xylene (2 × 5 min). Slides were rehydrated with successive baths of isopropanol (100%) and ethanol (100%, 96%, 70% and 50%) for 5 min each. Antigen retrieval was performed using a decloaking chamber (BioCare Medical, Concord, CA, USA) by heating the section in H_2_O at 110 °C for 20 min. Trypsin solution (25 µg/mL, 20 mM ammonium bicarbonate, and 0.01% glycerol) was deposited onto tissue sections using an automatic sprayer (15 µL/min, 16 cycles, TM Sprayer, HTX Technologies, Chapel Hill, NC, USA). The incubation time for digestion was 3 h at 50 °C in a wet chamber with a saturated solution of K_2_SO_4_. A CHCA (α-Cyano-4-hydroxycinnamic acid, Sigma Aldrich, St Quentin Fallavier, France) MALDI matrix (10 mg/mL in Acetonitrile/H_2_O, 70/30 *v*/*v*, 1% TFA (Trifloroacetique acid)) application step was also done with a TM sprayer (120 µL/min, 4 cycles). The MALDI-MSI analysis was performed in positive ionization with the reflectron mode, using the Smartbeam laser of the Autoflex III MALDI-TOF/TOF mass spectrometer. Data acquisition was performed using FlexControl 3.4 and FlexImaging 4.1 software packages (Bruker Daltonics, Bremen, Germany) in the range of *m*/*z* 600–3200 at spatial resolution of 100 μm. A peptide calibration standard mix, including angiotensin II, angiotensin I, substance P, bombesin, ACTH clip 1–17, ACTH clip 18–39, and somatostatin 28 (Bruker Daltonik GmbH), was used for external calibration. After MSI analysis, the MALDI matrix was removed from each slide by washing them in 100% ethanol for 5 min. Tissue sections were then stained with HE and examined by a pathologist to correlate MALDI data with histological features of the same section.

### 2.3. Data Processing

Data analysis of MALDI imaging data was performed with SCiLS Lab Pro 2023 (Bremen, Germany) on regions of interest corresponding to the different morphological patterns of cHCC-CCA, in order to assess the biomarkers of each region. The raw images were loaded into the SCiLS Lab software, and a baseline subtraction was performed using the convolution algorithm as well as a weak denoising. The spectra were normalized by total ion current (TIC). The peaks from selected regions were then aligned using a specific tool provided by Bruker [15].

Segmentation was performed using the software pipeline with weak denoising for the 5 cHCC-CCA cases. Relative quantification was performed for iCCA and HCC components, using the spectra number for specific clusters compared to the total count of spectra per case.

PCA analysis was achieved based on the segmentation pipeline peak list, working on all individual spectra with weak denoising and unit variance scaling.

In silico trypsin digestion of known immunohistochemical markers was performed on PeptideMass [16,17,18] (available online) with no protein treatment, and 1 missed cleavage and 2 unique peptides were selected using the neXtProt peptide uniqueness checker [19].

Moreover, protein annotation was assessed using the online version of Mascot with 1 missed cleavage allowed.

### 2.4. Informed Consent and Ethical Approval

All patients gave a written informed consent. The study was approved by our Ethical Committee “Comité d’Ethique de la Recherche (CER) Paris Nord” (Institutional Review Board -IRB 00006477- of HUPNVS, Paris 7 University, AP-HP): research N° CER-2022-169. It complies with the guidelines for human studies and was conducted ethically in accordance with the World Medical Association Declaration of Helsinki.

## 3. Results

### 3.1. cHCC-CCA Patients

The cHCC-CCA cases were obtained from surgical specimens from five patients [three men and two women, mean age 53.6 years (SD = 17.2)]. The main clinicopathological features are detailed in Table 1. The mean size of the tumors was 42.4 mm (SD = 15). Non-tumoral liver samples displayed advanced liver fibrosis (≥ F3 according to Metavir) in three patients (60%).

### 3.2. Segmentation Analysis Based on the MALDI-MSI Data

First, to evaluate the power of the MALDI-MSI analysis, segmentation was achieved on the pooled MSI data acquired on the five cases. The clustering analysis identified two main groups of spectra, which overlapped with the hepatocellular component (HCC and non-tumoral liver) and iCCA areas on the HE slides, respectively (Figure 1). In addition, tiny iCCA foci, not initially identified by histology, were observed within the HCC component in Cases 1 and 4, while small HCC areas were identified within iCCA regions in Case 2 (Figure 1, 1.1, 1.2 and 1.4 arrows). Furthermore, the segmentation analysis identified the presence of both tumor components (HCC and iCCA) within a tumor embol (Figure 1, 1.2, arrow).

Then, using segmentation analysis of the five cHCC-CCA cases, the relative proportions of the spectra belonging to either hepatocytic or cholangiocytic components were determined (Table 2). Tumor areas analyzed by MALDI imaging were more hepatocytic for Patient 3 (53.5%) and cholangiocytic for Patients 1, 2, 4, and 5 (71.4%, 70.6, 70.8, and 83.6, respectively) (Table 2).

Afterwards, segmentation was performed on these five cases individually. When opening up the two main clusters obtained, associated with HCC and iCCA components, respectively, MALDI imaging data provide further insights into cHCC-CCA heterogeneity. Compared to pathological annotations (Figure 2A), all spectra grouped under the green and blue areas were overlapped with an iCCA component (Figure 2B). More specifically, blue areas corresponded to fibrotic components in iCCA. On the other hand, the spectra grouped under the red and yellow colors were overlaid on an hepatocytic component. The yellow area specifically corresponded to fibrotic components in an hepatocytic area, including tumor capsules and fibrotic tissue in a non-tumoral liver delineating cirrhotic nodules (Figure 1, 1.2, and 1.4).

### 3.3. Analysis by Principal Component Analysis

In parallel to the segmentation study, a principal component analysis was performed. Similar results were found showing a clear distinction between iCCA and HCC areas (Figure 3A1). According to the first component, the two areas were highlighted in the five cases (Figure 3B) with a variance of 48.17% (Figure 3A3). Moreover, PCA analysis identified specific peaks belonging to the iCCA or HCC components, as shown in the loading plot according to Component 1 (Figure 3A2).

### 3.4. MALDI-MSI Segmentation and IHC Analysis

In order to confirm the ability of the MALDI imaging technique to capture the heterogeneity of cHCC-CCA, comparisons were carried out between the immunophenotypical markers routinely used for their diagnosis and the segmentation obtained by MSI; an example of the comparison is shown in Case 2 (Figure 4). In accordance with pathological annotation (Figure 4A), the group of spectra corresponding to the area of iCCA and the group of spectra corresponding to iCCA-derived fibrosis have been overlaid on the CK7 IHC. Areas positive for the CK7 marker were corresponding to the iCCA spectra group but not to the area with spectra derived from iCCA fibrosis (Figure 4C1 and Figure 4C2, respectively). On the other hand, areas associated with the hepatocytic or fibrotic component, including the tumor capsule and fibrosis in the non-tumoral liver, correctly overlapped with Glypican-3 (Figure 4C3) and HepPar-1 staining, respectively (Figure 4C4). In addition, segmentation identified areas of both tumor types (HCC and CCA) within a microvascular invasion (MVI) (Figure 4C3,C4).

### 3.5. Comparison between *In silico* Digestion and IHC Analysis

Since MALDI segmentation showed encouraging adequacy with IHC staining, we aimed to develop further MSI possibilities, such as obtaining virtual IHC stainings by using MALDI-MSI peptides. So, in order to generate a list of peptides comparable to those obtained following trypsin digestion of MALDI imaging samples, five immunohistochemical markers used for the routine diagnosis of cHCC-CCA [glypican-3 (P51654), CK 7 (P08729), EpCAM (P16422), CK 19 (P08727), and Nestin (P48681)] were digested in silico, using the PeptideMass online tool [16]. We did not use HepPar-1 because the commercial antibody used did not precisely recognize the target protein, and therefore the exact sequence could not be found for in silico digestion. Each peak list derived from the individual immunophenotypical markers was then filtered by overlaying the ionic peptide images on the histological scan. Two unique peptides per marker were selected and validated using the neXtProt peptide uniqueness checker (Appendix A). Then, the ionic images were compared with conventional IHC (Figure 5, reporting case 3). Images of the four other cases are shown in Appendix A.

### 3.6. Identification of Proteins of Interest

A Mascot search using the peak list obtained from the five cHCC-CCA cases was done to identify a list of proteins with a positive score (>56) (Appendix A). After a careful search of the available literature on data describing association with mechanisms of carcinogenesis or tumor progression, the proteins overexpressed in the HCC areas of greatest interest were Ladinin-1 (LAD1), 5-aminolevulinate synthase, nonspecific (ALAS1), mitochondria-eating protein (MIEAP), Protein FAM76B, Inactive phospholipase C-like protein 2 (PLCL2), Zinc finger protein 185 (ZN185), and Zinc finger protein 783 (ZN783). In CCA areas, Cytoplasmic dynein 1 heavy chain (DYHC1), DNA mismatch repair protein Mlh3, and Alpha-protein kinase 3 (ALPK3) were found more expressed. Finally, the peptide list suggested the presence of increased Kinectin 1, Coronin 1, and ALPK3 expression in both tumor types (HCC and iCCA).

## 4. Discussion

This study aimed to explore the intratumor heterogeneity of cHCC-CCA using MSI, a non-targeted imaging molecular approach. Our segmentation analysis based on peptidic profiles allowed the identification of two main clusters, corresponding to HCC and iCCA components, respectively. In addition, using MSI, small tumor foci initially not identified by routine histological analysis, were highlighted, suggesting the high potential of imaging to provide an exhaustive characterization of tumor tissue, compared to conventional histology. By definition, diagnosis of cHCC-CCA is histologically based in the identification of both HCC and CCA components, whatever their respective extent. However, the percentage of each component (HCC and iCCA) could be of main importance in predicting prognosis and guiding therapeutic decisions since treatment strategies are radically different between iCCA and HCC. Then, the varying percentages of composition of both tumor types would probably impact treatment response [1,4]. Interestingly, unsupervised analysis (as segmentation) from MSI could provide, in a quick and objective manner, an accurate proportion of each tumor type. Moreover, MALDI analysis, exploring the different clusters in a deeper way, has shown how intratumoral heterogeneity can be comprehensively studied. The impact of heterogeneity is not merely descriptive, as recently shown in breast cancers, in which the presence of high tumor heterogeneity has been associated with reduced survival [14].

Immunohistochemistry is not formally required for the diagnosis of cHCC-CCA but is always performed as an additional analysis to confirm the diagnosis and better phenotype the tumor. Using the list of peptides derived from in silico digestion, our method could reproduce and improve, on a single unlabeled tissue slide acquired by MSI, the information provided by five immunostainings derived from five successive tissue slides that may display substantial morphological differences. Our study is one of the first to use in silico digestion to derive markers from already-known immunophenotypical markers. Recently, a similar approach has been applied in breast cancers. This approach was used in a complementary way to multiplexed immunohistochemistry. MALDI techniques showed great potential but required several previous sample preparation steps that are for now expensive and time-consuming [20]. Although widely applied, inexpensive, and easy to interpret, immunohistochemical analyses have the inherent limitation of recognizing only a single protein. Our approach would theoretically allow unlimited evaluation of a large panel of markers from one single unstained slide. Indeed, markers for which routine IHC is not available could be sought by knowing the structure of the protein. However, it must be ensured that peptides analyzed by MSI technique are specific to the protein. Also, more precise characterization could be achieved using MS/MS analysis for further characterization of the targeted peptides. With our approach, MALDI imaging showed its full capacity to substitute five immunophenotypical markers for the accurate detection of iCCA and HCC components within the cHCC-CCA tumors.

Other mass spectrometry techniques, such as MS Probe Electrospray Ionization (PESI) combined with artificial intelligence, have already been used with good results for the identification of HCC and iCCA compared with healthy livers in patients undergoing liver resection [21,22]. This technique, in contrast to MALDI imaging, was based on solution analysis of the tissue area selected by the pathologist.

Finally, the list of peaks generated by MALDI analysis was used to identify proteins of interest using the Mascot database. Interestingly, proteins with high identification scores were identified. Some of the proteins found were already known in the literature as predisposing factors for various types of cancer, such as lung, breast, and gastrointestinal cancers. Noteworthy among them: overexpression of Ladinin-1 (LAD1), a known 59 kDa protein, has been associated with aggressive breast tumors and poor prognosis [23]. Nonspecific 5-aminolevulinate synthase (ALAS1) is another interesting protein, already associated with the development of various types of cancer, such as non-small-cell lung cancer and colorectal neoplasms [24,25]. At last, the cytoplasmic dynein heavy chain 1 (DYHC1), overexpresssed in CCA areas, has been associated with a high degree of epithelial-to-mesenchymal transmission (EMT) and a worse prognosis in hepatocellular carcinoma [26], and it has already been associated with the development of gallbladder tumors [27].

It remains well understood that this list of proteins of interest remains theoretical and has limited value, as it is not derived from identification techniques such as, for example, MS/MS. Therefore, we report our analysis and biological correlations with cHCC-CCA as an example of the multiple functionalities of a MALDI imaging analysis.

As an exploratory study, several limitations have to be noted. First, the small number of tumor cases included and the absence of controls. Second, limited access to the MALDI imaging approach, which requires a technical platform with a dedicated engineer in proteomics. Third, the complete analysis (from tissue section handling to data production and processing) remains relatively expensive compared to conventional analysis.

## 5. Conclusions

Our approach, although based on five cHCC-CCA cases, supports the potential of MALDI imaging for exploring intratumor heterogeneity at a deeper level compared to histology. We seek to exploit the full potential of the technique: on the one hand, the potential of unsupervised analysis, and, on the other hand, the ability to discover new peptidic markers through an in silico digestion approach. Further analysis should be carried out with more cases of cHCC-CCA and iCCA and HCC controls in order to validate this aptitude on a large scale. As demonstrated in several published studies, a good degree of reproducibility of MALDI imaging capabilities is achievable across laboratories [28,29]. Therefore, this MSI approach would increase in the future the diagnostic power of histological analysis by offering virtual immunohistochemical analysis that could ensure better patient management.

## Figures and Tables

**Figure 1 cancers-15-02143-f001:**
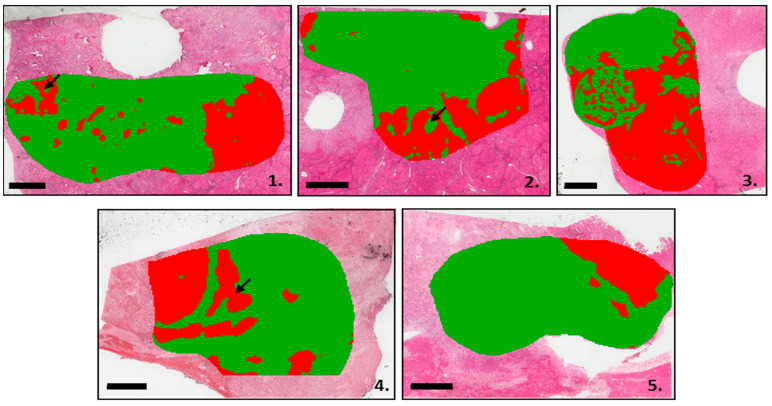
MALDI imaging segmentation analysis. The first arborescence of the segmentation analysis is shown here. Tissues with an hepatocyte component are highlighted in red, while tissues with an iCCA component are shown in green. HCC foci within an iCCA component were indicated by an arrow (1.1, 1.4). The iCCA foci within an HCC component were indicated by an arrow (1.2). Number 1 to 5 correspond to the five cases included. Scale bar = 3 mm.

**Figure 2 cancers-15-02143-f002:**
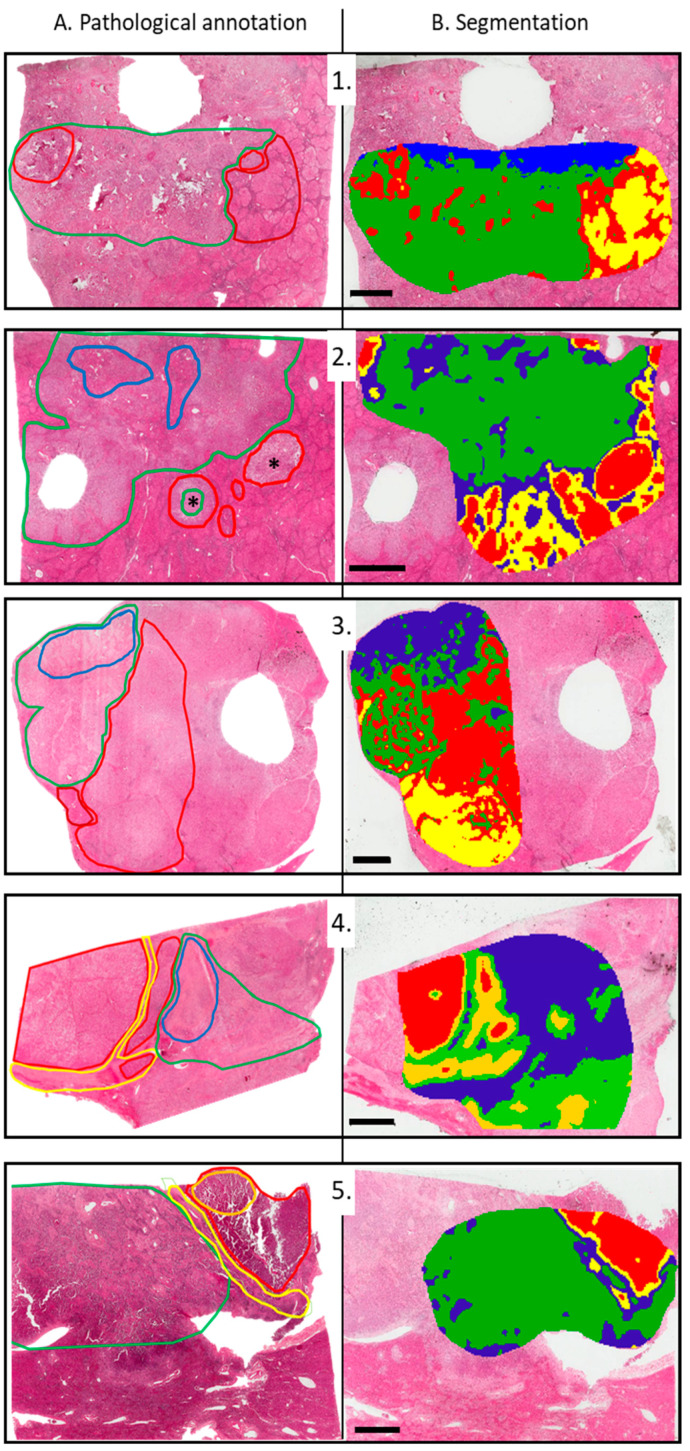
Pathological annotation compared to MALDI-MSI segmentation for the 5 cHCC-CCA cases. (**A**) Pathological annotations on HES obtained by two experienced pathologists following re-reading. The different colors indicate the areas of interest. Color legend: green: areas of cholangiocarcinoma; bright red: areas of hepatocarcinoma; blue: fibrosis derived from cholangiocarcinoma; yellow: fibrosis derived from hepatocarcinoma; orange: macrotrabecular area in hepatocarcinoma; deep red: non-tumoral liver. Two asterisks: MVI. (**B**) MALDI-MSI segmentation obtained separately and overlaid with successive HES slices. Color corresponds to similar spectra groups: red: HCC; yellow: fibrotic component including capsule of HCC and fibrosis around cirrhotic nodules; green: iCCA; blue: fibrotic component in iCCA. Number 1 to 5 correspond to the five cases included. Scale bar = 3 mm.

**Figure 3 cancers-15-02143-f003:**
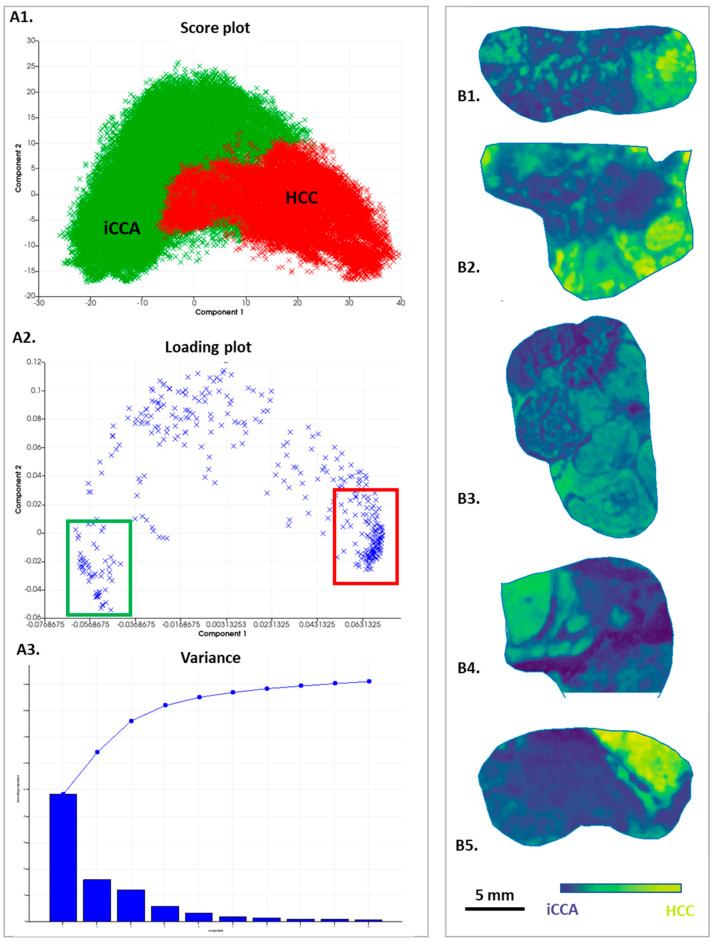
Principal component analysis of the 5 cHCC-CCA cases. (**A1**): Score plot with iCCA component annotated in green and HCC component annotated in red. (**A2**): Loading plot of PCA with specific peptides of the 2 components boxed in green (iCCA) and red (HCC). (**A3**): Variance of the PCA analysis. (**B1**–**B5**): Images of the 5 cases corresponding to the first component within the deep blue iCCA component and in the yellow HCC component.

**Figure 4 cancers-15-02143-f004:**
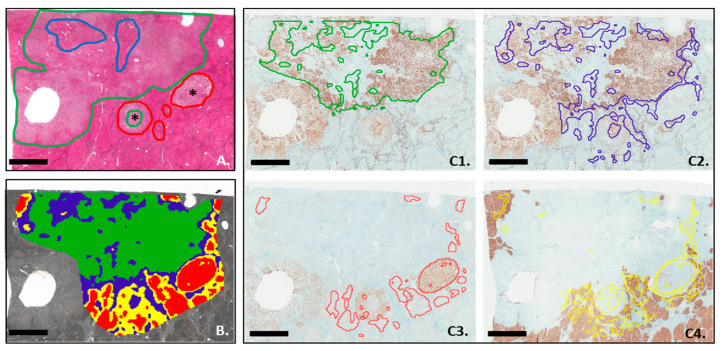
Segmentation analysis overlaid with immunohistochemistry performed on successive slices from Patient 2. (**A**) Pathological annotations with colors that indicated areas of cholangiocarcinoma (green), areas of hepatocarcinoma (red), fibrosis derived from cholangiocarcinoma (blue), and fibrosis from a tumor capsule (yellow). Asterisks: MVI. (**B**) The segmentation analysis was set to four clusters: blue: cholangiocarcinoma-related fibrosis; red: HCC; green: iCCA; and yellow: fibrosis around cirrhotic nodules. (**C1**,**C2**) = CK7 staining; (**C3**) = glypican staining; (**C4**) = HepPar-1 staining. Scale bar = 3 mm.

**Figure 5 cancers-15-02143-f005:**
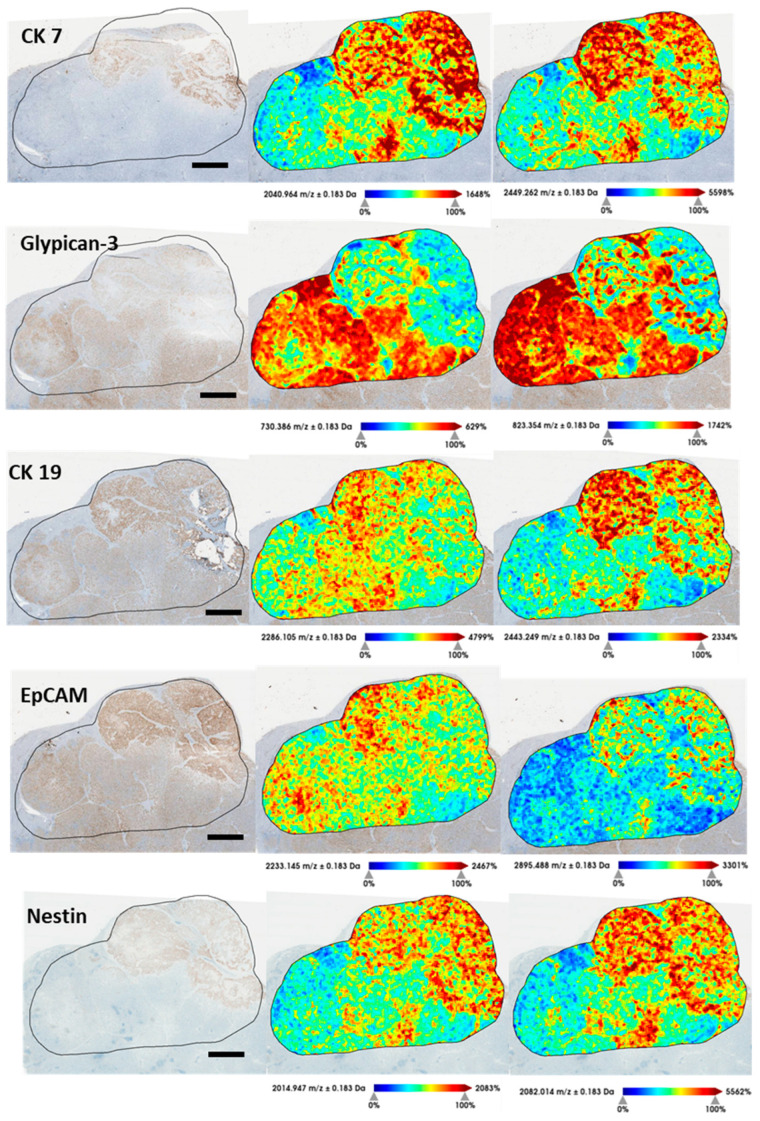
Comparison of a selection of tryptic peptides derived from current IHC markers and the corresponding IHC for Patient 3. Scale bar = 3 mm. It can be seen that there is good overlap between the CK7-labeled zones and the peptides derived from in silico digestion of the same protein. It can also be seen that tryptic peptide is detected in areas not labeled by CK7; these areas, upon careful morphological analysis, had areas of cholangiocyte differentiation. Similarly, good overlap can be seen between the markers glypican-3, Nestin, EpCAM, and CK19 with related tryptic peptides. Cases 1, 2, 4, and 5 are reported in Appendix A.

**Table 1 cancers-15-02143-t001:** Clinical and pathological features of patients.

Sex	Age (Y)	Advanced Fibrosis (≥F3)	Risk Factors	BMI	Tobacco Smoke	Satellite Nodules	Vascular Invasion	Perineural Invasion	Tumor Size (mm)
F	65	Yes	Alcohol, MS	40	Yes	Yes	Yes	Yes	42
M	57	Yes	Alcohol, MS	27	Yes	No	Yes	Yes	25
F	29	Yes	Genetic	44	Yes	No	No	No	35
M	40	No	HBV	31	Yes	No	Yes	Yes	40
M	77	No	MS	24	Yes	Yes	Yes	Yes	70

MS: Metabolic syndrome, HBV: hepatitis B virus, BMI: Body Mass Index.

**Table 2 cancers-15-02143-t002:** Quantification of hepatocyte and cholangiocyte components using MALDI-MSI segmentation.

		iCCA (Green)		HCC (Red)	
Patients	Total of Spectra	Number of Spectra	%	Number of Spectra	%
1	20,438	14,602	71.4	5836	28.6
2	14,968	10,549	70.5	4419	29.5
3	20,650	9612	46.5	11038	53.5
4	16,360	11,584	70.8	4776	29.2
5	14,158	11,843	83.6	2315	16.4

## Data Availability

All data generated or analyzed during this study are included in this article [and/or] its Appendix A. Further enquiries can be directed to the corresponding author.

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
