# Peer review of "MALDI Imaging, a Powerful Multiplex Approach to Decipher Intratumoral Heterogeneity: Combined Hepato-Cholangiocarcinomas as Proof of Concept"

_cancers, 2023, doi:10.3390/cancers15072143_

Round 1

Reviewer 1 Report

1. Please mention more about diagnosis of cHCC-CCA based on the percentage of two components(HCC and CCA) if possible.

2. MALDI imaging method is not a routine method for identification for pathological diagnosis, what difference between traditional pathological diagnosis and your method.

Author Response

Reviewer # 1

  1. Please mention more about diagnosis of cHCC-CCA based on the percentage of two components (HCC and CCA) if possible.

Answer:

We thank reviewer #1 for this comment, which is a main issue in the field of hepatocholangiocarcinomas, the definition of which has indeed changed over time. According to the last WHO classification, cHCC-CCA is defined by the presence of both components (HCC and iCCA) whatever their respective proportion. In the initial manuscript (section discussion), a sentence detailed that point “By definition, diagnosis of is histologically based with the identification of both HCC and CCA components, whatever their respective extent. However, the percentage of each component (HCC and iCCA) could be of main importance in predicting prognosis and guiding therapeutic decisions since treatment strategies are radically different between iCCA and HCC”.

To emphasize this point, we added in the introduction section the following sentence with the appropriate reference: “Also, it is important to note that cHCC-CCA diagnosis is proposed whatever their respective proportion. “

  1. MALDI imaging method is not a routine method for identification for pathological diagnosis, what difference between traditional pathological diagnosis and your method.

Answer:

MALDI imaging is an in situ approach providing a spatialized molecular picture based on expression of various kinds of molecules (proteins, peptides, lipids and metabolites) directly from tissue sections without preliminary microdissection and labelling steps. In the current study, we investigated protein profiles of cHCC-CCA cases by MALDI imaging, which is, as outlined by the reviewer, still considered as a research approach mainly used for identifying new molecular tissue biomarkers. Then, traditional pathological diagnosis of cHCC-CCA is routinely made by conventional histological analysis based on hematoxylin and eosin staining, allowing to identify tumor components (HCC and CCA) according to cytological and architectural features. Such morphological diagnosis is routinely supported in most of the cases by additional immunohistochemical analysis, which highlights the 2 components according to immunophenotypical features.

This point has been introduced in the initial paper as follows:

“The latest classification has been simplified and recognizes as cHCC-CCA only those lesions in which HCC and CCA areas are present based on routine hematin-eosin (HE) staining. Also, it is important to note that cHCC-CCA diagnosis is proposed whatever their respective proportion.  To support the morphological diagnosis, some additional immunohistochemical (IHC) markers reflecting cell differentiation (hepatocytic and cholangiocytic) as well as stem cell phenotype are used; these latter are useful to better characterize PLC and to guide differential diagnoses without being diagnostic by themselves.”

Our objective to use MALDI imaging was to explore at best intra-tumor heterogeneity at the molecular level through unsupervised analysis. In addition, MALDI imaging allowed, through in silico digestion techniques, to generate a kind of "virtual immunohistochemistry" painting various markers from one single slide acquired by MSI.

Reviewer 2 Report

The article gives a MALDI-MSI analytical investigation on the heterogeneity of combination hepatocellular-cholangiocarcinoma (cHCC-CCA). MALDI imaging was able to detect the existence of both tumour components (HCC and iCCA) inside a tumour embol and gave further insight into cHCC-CCA heterogeneity, according to the research. Also, the findings of principal component analysis and a comparison between MALDI-MSI segmentation and immunohistochemistry (IHC) analysis are discussed.

The following experiments may be performed for improving this article:

1-While the research included just five patients, a larger sample size would have helped validate the findings and extend them to a wider population.

2-It would be beneficial to compare the findings of MALDI-MSI with those of other imaging methods, such as computed tomography (CT) and magnetic resonance imaging (MRI), in order to assess the sensitivity and specificity of MALDI-MSI in detecting the different tumor components.

3-The paper does not evaluate the clinical relevance of the results. To determine the prognostic usefulness of the heterogeneity discovered by MALDI-MSI in cHCC-CCA patients, more research might be conducted.

 4-It would be interesting to compare the findings of MALDI-MSI with those of other proteomics methods, such as liquid chromatography-mass spectrometry (LC-MS), to assess the sensitivity, specificity, and repeatability of MALDI-MSI in detecting various tumour components.

Author Response

Reviewer # 2

The article gives a MALDI-MSI analytical investigation on the heterogeneity of combination hepatocellular-cholangiocarcinoma (cHCC-CCA). MALDI imaging was able to detect the existence of both tumour components (HCC and iCCA) inside a tumour embol and gave further insight into cHCC-CCA heterogeneity, according to the research. Also, the findings of principal component analysis and a comparison between MALDI-MSI segmentation and immunohistochemistry (IHC) analysis are discussed.

The following experiments may be performed for improving this article:

  1. While the research included just five patients, a larger sample size would have helped validate the findings and extend them to a wider population.

Answer:

We thank reviewer #2 for his comment.  We fully agree that a larger sample size would be required to validate some results from a clinical point of view. Thus, we had outlined this limitation in the discussion section as follows: "Further analysis should be carried out including more cases of cHCC-CCA and iCCA and HCC controls in order to validate this aptitude on a large scale."

We would like to stress that our main objective was to conduct an exploratory study to evaluate whether MALDI imaging could be useful to decipher intratumor heterogeneity of cHCC-CCA, a rare entity of primary liver malignancies, for which number of cases available is very limited, To address our issue, we therefore selected, among our archival pathology files, typical and representative cases with various proportions of tumor components. Based on our positive results, we plan to design a validation study including a great number of cases from our center and other French centers. To do so, and reduce the number of cases to be subjected to MALDI imaging, we will build tissue micro-arrays composed of 8 representative tumor areas per case. This collaborative project will allow to study the potential impact on prognosis.

  1. It would be beneficial to compare the findings of MALDI-MSI with those of other imaging methods, such as computed tomography (CT) and magnetic resonance imaging (MRI), in order to assess the sensitivity and specificity of MALDI-MSI in detecting the different tumor components.

Answer:

Unfortunately, as we described in a previous article (Gigante et Al. Liver International 2019) the performance of radiology alone in the diagnosis of these tumors is poor (48% sensitivity and 81% specificity. We also reported that the best performance for cHCC-CCA diagnosis is obtained by the following 2 step strategy 1-radiology and 2-biopsy analysis (60% sensitivity, 82% specificity). In line with these published data, we strongly believe that the diagnosis of cHCC-CCA would be reached at best by including additional morphological and molecular features (such as obtained by MALDI imaging).

Again, our main objective was not to demonstrate the performance of MALDI imaging for cHCC-CCA diagnosis, but to illustrate and better understand intratumor heterogeneity (at the mm scale) in this type of tumor. The macroscopic radiological heterogeneity (at the cm scale) of cHCC-CCA is indeed a very interesting subject that has been poorly evaluated (rare liver tumors).

  1. The paper does not evaluate the clinical relevance of the results. To determine the prognostic usefulness of the heterogeneity discovered by MALDI-MSI in cHCC-CCA patients, more research might be conducted.

Answer:

As already answered (see point 1 Rev 2), the purpose of our study was not to validate nor evaluate the clinical or prognostic impact of MALDI technique. The limited number of patients included limits the clinical conclusions we can make. A study with a larger number of patients, the presence of controls and clinical correlations is currently designed.

  1. It would be interesting to compare the findings of MALDI-MSI with those of other proteomics methods, such as liquid chromatography-mass spectrometry (LC-MS), to assess the sensitivity, specificity, and repeatability of MALDI-MSI in detecting various tumour components.

Answer:

We fully agree with the comment stating that additional methods would be useful to validate results obtained by MALDI imaging. Such study, requiring careful microdissection of tumor areas (that as shown in the paper not always identified by the conventional histological analysis) will be performed in the future.

Reviewer 3 Report

I have read with interest this manuscript that concerns the use of a MALDI procedure to define and characterize 5 cases of combined HCC-CCA. I congratulate the authors for their very preliminary results, and I would suggest the following changes.

1. Please discuss your techniques and studies in the light of some recent and similar studies (Giordano et al, Liver Int 2020; Giordano et al. Applied Sciences 2022).

2. Did you tested your technique for the indentification of some important histological prognosti factors such as, for instance, microvascular invasion? Can your technique anticipate such factor?

3. Where the analyses performed blind to the pathologist report?

4. Even if this is a proof of concept, please moderate your conclusions that are currently based only on 5 patients. 

5. The section of study limitations is missing.

6. Please add a paragraph to detail the cost (approximative) and the time required for this analysis. 

Author Response

Reviewer # 3

I have read with interest this manuscript that concerns the use of a MALDI procedure to define and characterize 5 cases of combined HCC-CCA. I congratulate the authors for their very preliminary results, and I would suggest the following changes.

  1. Please discuss your techniques and studies in the light of some recent and similar studies (Giordano et al, Liver Int 2020; Giordano et al. Applied Sciences 2022).

Answer:

We thank the reviewer #3 for his/her advice. We have included a sentence in the discussion to introduce the two studies with related references:

“Other mass spectrometry techniques, such as MS Probe Electrospray Ionization (PESI) combined with artificial intelligence, have already been used with good results for the identification of HCC and iCCA compared with healthy liver in patients undergoing liver resection REF REF. This technique, by contrast to MALDI imaging, was based on solution analysis of the tissue area selected by the pathologist.”

  1. Did you tested your technique for the indentification of some important histological prognosti factors such as, for instance, microvascular invasion? Can your technique anticipate such factor?

Answer:

To note, we were able to identify using MALDI imaging surrogate markers of microvascular invasion in HCC in a previous study (Poté et al Hepatology 2013), showing the performance of this approach in such field. While the objective of the current exploratory study, as previously discussed, was not to identify specific markers, we are very confident that prognostic tissue biomarkers could be found. This is strongly supported by the identification of specific molecular pattern we observed in areas corresponding to vascular emboli.

  1. Where the analyses performed blind to the pathologist report?

Answer:

No, the pathologist, aware of the diagnosis, performed a careful histological analysis to retrieve representative cases of cHCC-CCA. In addition, the pathologist  selected the tumor slide that includes both HCC and iCCA components.

  1. Even if this is a proof of concept, please moderate your conclusions that are currently based only on patients.

Answer:

We fully understand reviewer’s comment, and modified the conclusions inserting the following sentences:

“Our approach, although based on 5 cHCC-CCA cases, supports the potential of MALDI imaging for exploring intratumor heterogeneity at a deeper level compared to histology”

And

“Therefore, this MSI approach, would increase in the future the diagnostic power of histological analysis by offering virtual immunohistochemical analysis that could ensure better patient management.”

  1. The section of study limitations is missing.

See below

  1. Please add a paragraph to detail the cost (approximative) and the time required for this analysis. 

Answer:

In response to points 5 and 6, we included a small paragraph in the discussion regarding the limitations of our study.

“As an exploratory study, several limitations have to be raised. First, the small number of tumor cases included and the absence of controls. Second, the limited access of MALDI imaging approach, which requires a technical platform with dedicated engineer in proteomics. Third, the complete analysis (from tissue section handling to data production and processing) remains relatively expensive compared to conventional analysis.”

Round 2

Reviewer 2 Report

I am satisfied with the author's responses.